# Life and death of a leprosy sufferer from the 8th-century-CE cemetery of Kiskundorozsma–Kettőshatár I (Duna-Tisza Interfluve, Hungary)—Biological and social consequences of having Hansen's disease in a late Avar Age population from Hungary

Olga Spekker[1,2]*, Balázs Tihanyi[1,3], Luca Kis[1,3], Csaba Szalontai[4], Tivadar Vida[2], György Pálfi[1], Antónia Marcsik[1], Erika Molnár[1]

1 Department of Biological Anthropology, University of Szeged, Szeged, Hungary, 2 Institute of Archaeological Sciences, Eötvös Loránd University, Budapest, Hungary, 3 Department of Archaeogenetics, Institute of Hungarian Research, Budapest, Hungary, 4 Archaeological Heritage Protection Directorate, Hungarian National Museum, Budapest, Hungary

* olga.spekker@gmail.com

## Abstract

The aim of our paper is to demonstrate a middle-aged male (**KK61**) from the 8th-century-CE cemetery of Kiskundorozsma–Kettőshatár I (Duna-Tisza Interfluve, Hungary), who appears to represent the lepromatous form of Hansen's disease. Leprosy has affected not only the rhinomaxillary region of his face but also his lower limbs, with severe deformation and disfigurement of the involved anatomical areas (saddle-nose and flat-foot deformity, respectively). Consequently, he would have experienced disability in performing the basic activities of daily living, such as eating, drinking, standing or walking; and thus, he would have required regular and substantial care from others to survive. Despite his very visible disease and associated debility, it seems that **KK61** was accepted as a member of the community in death, since he has been buried within the cemetery boundaries, among others from his community. In addition, his grave has conformed to the mortuary practices characteristic of the Kiskundorozsma–Kettőshatár I cemetery (e.g., burial orientation, position of the body in the grave, and type and quantity of accompanying grave goods). Although distinction or segregation in life do not preclude normative treatment in death, the long-lasting survival of **KK61** with Hansen's disease implies that he would not have been abandoned but cared for by others. **KK61** is one of the few published historic cases with leprosy from the Avar Age of the Hungarian Duna-Tisza Interfluve. His case gives us a unique insight into the biological consequences of living with Hansen's disease and illustrates the social attitude toward leprosy sufferers in early mediaeval Hungary.

**Data Availability Statement:** All relevant data are within the manuscript and its Supporting Information files.

**Funding:** This work was funded by the University of Szeged Open Access Fund (grant agreement n˚ 5530) to OS. The National Research, Development and Innovation Office (Hungary) (grant agreement n˚ K 125561) and the "Árpád-ház Program" (grant agreement n˚ 39509/2018/KFSZ) of the Hungarian Ministry of Human Capacities provided funding for GP. This project received funding from the European Research Council (ERC) under the European Union's Horizon 2020 research and innovation programme (grant agreement n˚ 856453 ERC-2019-SyG) to TV. The funders had no role in study design, data collection and analysis, decision to publish, or preparation of the manuscript.

**Competing interests:** The authors have declared that no competing interests exist.

# Introduction

## Leprosy

Leprosy or Hansen's disease (HD) is a chronic infection of mainly humans that is caused by two pathogenic bacterial species, *Mycobacterium leprae* and *Mycobacterium lepromatosis* [1, 2]. Leprosy bacilli are most commonly transmitted from one person to another through respiratory droplets, with the nasal mucosa representing the main portal of entry and exit for the pathogens [1, 3, 4]. The infection can also be spread via prolonged direct skin-to-skin contact with an afflicted individual or via zoonotic transmission to humans by nine-banded armadillos (*Dasypus novemcinctus*) [2, 4]. Nevertheless, the vast majority of those exposed to leprosy bacilli (over 90%) will never become infected and develop clinically active HD, as they are genetically not susceptible to it [2, 5].

Even if a person becomes infected, usually, there is an extended incubation period of approximately 5 years on average between the transmission of HD and the onset of its clinical signs and symptoms (asymptomatic latent infection) [1, 3]. Amongst the few individuals who do progress to symptomatic disease, leprosy is manifest over a broad clinical and histopathological spectrum from tuberculoid through borderline forms to lepromatous [1, 3, 6]. The presentation of the disease reflects the host's immune response to leprosy bacilli–patients have decreasing levels of cell-mediated immunity and increasing levels of humoral immunity as they move from the tuberculoid form to the lepromatous one [6, 7].

As the optimum growth temperature of leprosy bacilli is below the core body temperature, the disease primarily affects the superficial, and thereby cooler areas of the human body, such as the skin, the peripheral nerves within or close to the skin, and the nasal mucosa [1, 3]. Later, the skeleton can also become involved by HD either through direct extension of the infection from adjacent soft tissues or through haematogenous dissemination of leprosy bacilli from the primary site of infection into the bone [8, 9]. In addition, skeletal lesions can develop secondary to leprous peripheral neuropathy [8, 10]. Bony changes can occur at and between both ends of the disease spectrum, with the rhinomaxillary region of the face, the small bones of the hands and feet, and the long tubular bones of the arms and legs being the most important predilection sites [8, 10, 11].

HD is one of the few bacterial infections that can readily be recognised in ancient human remains based on characteristic macromorphological alterations to the skeleton resulting from the disease [12, 13]. Lepromatous and near-lepromatous leprosy are characterised by simultaneous involvement of the rhinomaxillary region and the postcranial elements, whereas tuberculoid and near-tuberculoid leprosy are characterised by involvement of the postcranial skeleton with no rhinomaxillary lesions [14–16]. It is important to bear in mind that the individual bony changes that can be associated with HD are not pathognomonic to the disease as other pathological conditions, such as syphilis and tuberculosis, or taphonomic processes can cause similar or even the same alterations [8, 10]. It is the specific distribution pattern of leprous lesions in different areas of the human skeleton rather than the single bony changes themselves that can provide a definitive diagnosis of the disease in archaeological materials [8, 17].

## Archaeological background

After the fall of the Roman Empire, there was a dramatic increase in the prevalence of leprosy in Europe during mediaeval times [18]. It was proposed that the successive westward migration of the nomadic Avar tribes from Central Asia or Asia Minor in the first millennium CE either first introduced or re-transmitted different *Mycobacterium leprae* strains into Eastern and Central Europe, including Hungary [18–20]. Furthermore, it would certainly have

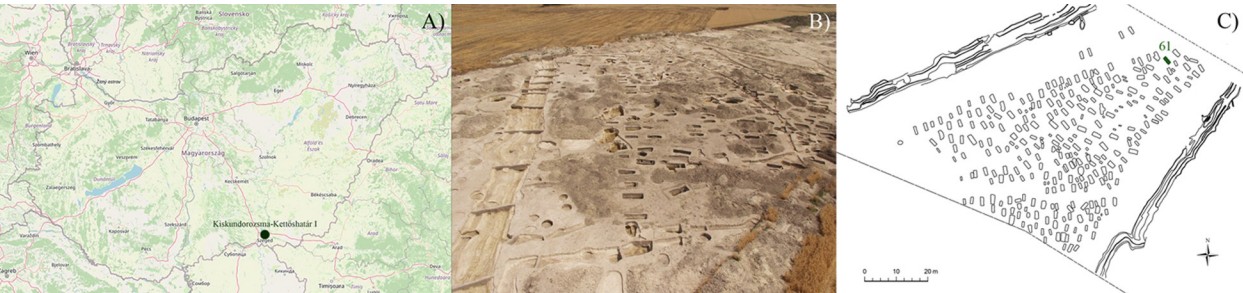

**Fig 1.** A) Map of Hungary showing the location of the Kiskundorozsma–Kettőshatár I archaeological site; B) Aerial photo of the Kiskundorozsma–Kettőshatár I archaeological site; and C) Plan drawing of the late Avar Age cemetery of Kiskundorozsma–Kettőshatár I with the location of the burial of KK61. (Fig 1A contains information from OpenStreetMap and OpenStreetMap Foundation, which is made available under the Open Database License).

facilitated the spread of any HD already existing in this geographical region during the early mediaeval period [18]. Palaeopathological and palaeomicrobiological studies have provided clear evidence for the presence of leprosy in past human populations from Avar Age Hungary but the number of published cases with HD, especially from the Duna-Tisza Interfluve of the country, is scarce [21, 22].

Some of the aforementioned cases derive from the Avar Age cemetery of Kiskundorozsma–Daruhalom-dűlő II [20, 21]. It is dated to the late 7[th] century CE (early/middle Avar transition period) and geographically located in close vicinity to the present-day town of Kiskundorozsma, at the western bank of the Maty Creek (Fig 1A) [23, 24]. About 200 m east of the Kiskundorozsma–Daruhalom-dűlő II burial site, at the eastern bank of the Maty Creek, remains of another Avar Age cemetery, Kiskundorozsma–Kettőshatár I, have been found in 2004 [24, 25]. It seems that the abandonment of the Kiskundorozsma–Daruhalom-dűlő II cemetery (around the 670s CE) may have coincided with the opening of the Kiskundorozsma–Kettőshatár I cemetery [24–26]. It cannot be excluded that the two burial sites, situated very close to each other but separated by the Maty Creek, were established by subsequent generations of the same Avar community [24–26].

Prior to and simultaneously with the construction works of motorway M43, a rescue excavation, directed by Patrícia Mészáros, Tibor Paluch, and Csaba Szalontai, was carried out at the archaeological site of Kiskundorozsma–Kettőshatár I (Fig 1B) [24, 27]. The excavation lasted from 26 April until 30 June 2004, and involved an area of 31,422 m$^2$ that coincided with the trail of motorway M43 [24, 27]. The Avar Age cemetery of Kiskundorozsma–Kettőshatár I was discovered at the western part of the large excavation area [24]. Although the burial site was enclosed by ditches (7 to 9 m in width) on its eastern and western sides, its exact extent is not known as its northern and southern boundaries laid outside the excavation area [24, 26]. Therefore, the cemetery can be considered as only partially excavated [24–26].

In the Kiskundorozsma–Kettőshatár I burial site, a total of 298, very densely distributed graves were uncovered, which makes it one of the largest Avar Age cemeteries of Csongrád-Csanád county, Hungary (Fig 1C) [24–27]. Along the western edge of the burial site, the graves were arranged in ordered rows, whereas towards the central area, they exhibited a more irregular distribution [24]. The grave pits, usually 0.4 to 1.8 m in depth, were of rectangular in shape with rounded corners, more or less vertical walls, and an even bottom [24, 26, 27]. In about one-third of the graves, there was evidence for the presence of a rectangular, wooden coffin with a length and width barely larger than the same dimensions of the deceased [24, 26]. The corpses were generally buried in an extended supine position and oriented from northwest to southeast, with

the head at the northwestern end of the grave [24, 26, 27]. In a number of cases, the body of the deceased may have been tightly wrapped in canvas before placed into the coffin [24].

Regardless of the age or sex of the deceased, about half of the burials contained animal bones, most commonly domesticated ruminants and poultry [24]. Among the items unearthed from the burials, the most common ones were iron knifes and buckles in male graves, and spindle buttons in female graves [24]. About one-fifth of the burials appeared to be disturbed as a result of contemporary grave robbery, especially at the northeastern part of the cemetery [24, 26]. In the majority of the robbed graves, it was the upper body that was targeted [24, 26]. Based on the associated grave goods, the cemetery of Kiskundorozsma–Kettőshatár I may have been in use from the end of the middle Avar Age (last third of the 7[th] century CE) until the turn from the 8[th] to the 9[th] century CE [24–26].

## Aims

The aim of our paper is to demonstrate an individual (**KK61**) from the 8[th]-century-CE cemetery of Kiskundorozsma–Kettőshatár I (Duna-Tisza Interfluve, Hungary), who appears to represent the lepromatous form of HD. The case of **KK61** has been only briefly summarised in a review by Marcsik and her colleagues [21] but the detailed macromorphological description of the bony changes observed in the skeleton has not yet been provided. To reconstruct the type of leprosy and the biological consequences of the disease progression in **KK61**, the detected leprous lesions were linked with palaeopathological and modern medical information. Furthermore, to reconstruct the social consequences of being afflicted with HD in the late Avar Age community of Kiskundorozsma–Kettőshatár I, conceptualisation of the examined individual's treatment in death was conducted. **KK61** is one of the few published historic cases with leprosy from the Avar Age of the Hungarian Duna-Tisza Interfluve. The case of **KK61** gives us a unique insight into the biological consequences of living with HD and illustrates the social attitude toward leprosy sufferers in early mediaeval Hungary.

## Materials and methods

### Burial context

The skeleton of **KK61** was unearthed from burial 61 of the Kiskundorozsma–Kettőshatár I cemetery (Fig 1C). In the northwest-southeast oriented, rectangular pit-grave, that had rounded corners, vertical walls, and an even bottom, the skeleton laid in an extended supine position (Fig 2A) [25]. The upper body was not disturbed, with the skull tilted to the right, the arms and forearms tucked closely alongside the body, and the hands resting on the outer thighs [25]. In the lower body, both legs were extended with the knees touching; the feet were slightly disturbed [25]. The grave goods found in burial 61 were the following: an iron object next to the left humerus, an iron buckle and knife on the right iliac wing, chicken bones next to the left lower leg, and pig bones [25]. Burial 61 can be dated to the last phase (late 8[th]–early 9[th] century CE) of the Kiskundorozsma–Kettőshatár I cemetery [25]. The skeletal remains of **KK61** (inventory no. AP473) are currently housed at the Department of Biological Anthropology, University of Szeged (Szeged, Hungary).

### Methods

The relatively complete and well-preserved skeleton of **KK61** (Fig 2B) was subject to a detailed macromorphological analysis, focusing on the detection of pathological bony changes probably related to leprosy. Prior to the palaeopathological examination of **KK61**, age-at-death was estimated and sex was determined applying standard macromorphological methods of

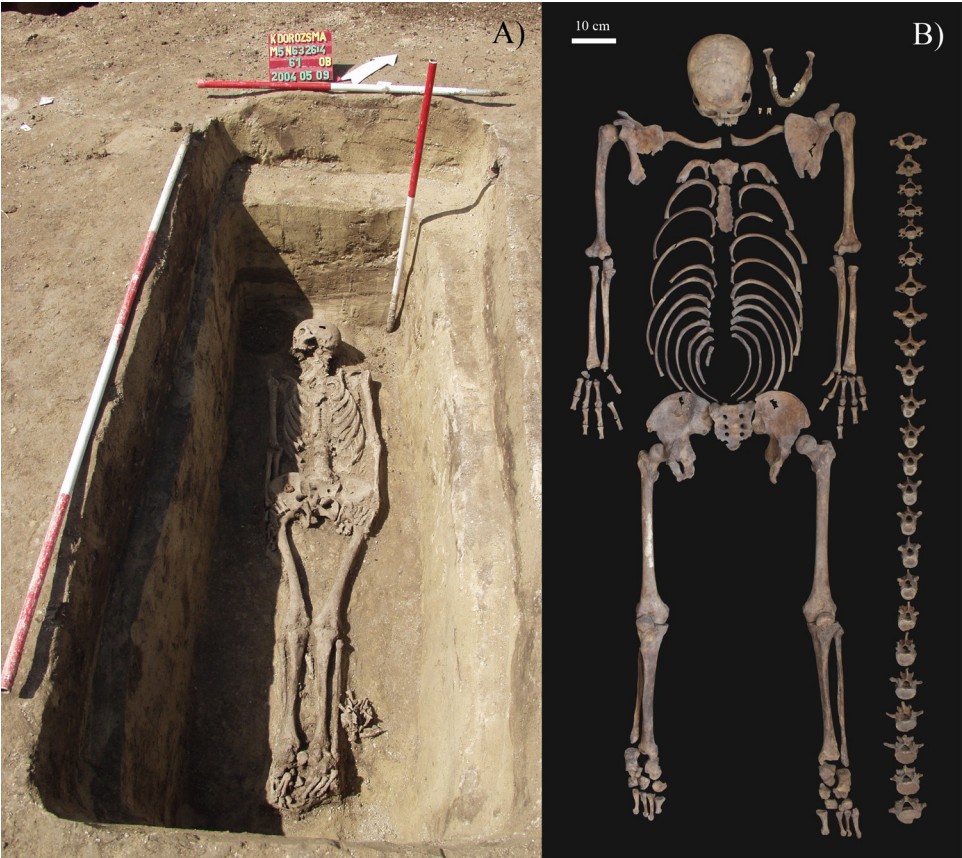

**Fig 2.** A) Photo of the burial of KK61 *in situ*; and B) Completeness of the skeleton of KK61.

bioarchaeology [28–31]. Based on the results of these investigations, the skeleton belonged to a middle-aged (c. 45–50 years old) male. During the palaeopathological evaluation, all skeletal remains of **KK61** were macroscopically examined with the naked eye. The diagnostic criteria for HD, that were investigated during the macromorphological analysis, can be seen in Table 1 [17, 32–40]. In addition to the aforementioned palaeopathological diagnostic criteria for leprosy, signs of maxillary sinusitis, *cirbra orbitalia*, and certain types of tooth pathologies (e.g., dental caries, alveolar bone recession, and dental calculus) were also evaluated as these alterations have been frequently observed in cases with HD [41–45].

**Ethics statement.**   Specimen number: **KK61** (inventory no. AP473; grave no. 61).

The skeleton evaluated in the described study is housed in the Department of Biological Anthropology, University of Szeged, in Szeged, Hungary. Access to the specimen was granted by the Department of Biological Anthropology, University of Szeged (Közép fasor 52, H-6726 Szeged, Hungary).

No permits were required for the described study, which complied with all relevant regulations.

## Results

### Cranial bony changes

The rhinomaxillary region of **KK61** exhibited a number of alterations that can presumably be attributed to leprosy (Fig 3). The anterior nasal spine (Fig 4A, 4B and 4D) completely

**Table 1. Macromorphological diagnostic criteria for leprosy.**

| | Diagnostic criteria for leprosy | References |
|---|---|---|
| **Rhinomaxillary region of the face** | Surface pitting, progressive resorption, and eventual disappearance of the anterior nasal spine, with subsequent cortical capping at its original base | [32, 33, 36] |
| | Progressive, bilaterally symmetrical resorption and rounding/remodelling of the inferior and lateral margins of the pyriform aperture, with inferior widening of the anterior bony opening of the nasal cavity | [32, 33, 36] |
| | Progressive resorption, recession, and remodelling of the maxillary alveolar process (restricted largely to the premaxilla), with loosening and ultimate *ante-mortem* loss of the maxillary incisors | [32, 33, 36] |
| | Surface pitting, erosion, and thinning of the nasal and/or oral surfaces of the maxillary palatine process (sometimes accompanied by subperiosteal new bone formation), with eventual perforation of the hard palate | [32, 33, 36] |
| | Surface pitting, progressive resorption, and ultimate disappearance of the intranasal bony structures, especially the bony nasal septum and the inferior nasal conchae | [32, 33, 36] |
| **Postcranial skeleton** | Surface pitting, subperiosteal new bone formations, and/or exostoses on the shaft of the lower leg bones, especially the distal half or two-thirds | [17, 38–40] |
| | Palmar grooving at the distal end of the proximal hand phalanges | [34] |
| | Palmar bevelling at the proximal end of the middle hand phalanges | [34] |
| | Exostoses on the dorsal surface of the tarsal bones | [35] |
| | Concentric diaphyseal atrophy of the metacarpals, metatarsals, hand and/or foot phalanges, with or without accompanying achro-osteolysis | [37] |
| | Septic bony changes of the small bones of the hands and/or feet (e.g., surface pitting, subperiosteal new bone formation, cortical erosion, and lytic or cystic lesions) | [38] |
| | Septic articular changes of the joints of the hands and/or feet (e.g., subluxation, dislocation, and bony ankylosis) | [38] |

disappeared. At its original base, there were signs of cortical capping (Fig 4A, 4B and 4D). On both sides, the pyriform aperture (Fig 4A, 4B and 4D), particularly its inferior half, was widened and rounded with resorption and remodelling of its margins. At the prosthion, there was slight resorption of the maxillary alveolar process (Fig 4B–4D). The alveoli of the maxillary central incisors were severely destroyed, whereas the alveoli of the maxillary lateral incisors were only slightly damaged (Fig 4B–4D). Both the oral (Fig 4C) and nasal (Fig 4D) surfaces of the maxillary palatine process displayed extensive pitting, erosion, and thinning, maximal towards the median palatine suture. There was a large, irregular, sharp-edged perforation at the posterior half of the hard palate (Fig 4C and 4D). Both inferior nasal conchae (Fig 4A, 4B and 4D) were entirely resorbed. Although it was slightly damaged *post-mortem*, it seems that the bony nasal septum (Fig 4B and 4D) was at least partially (inferiorly) absorbed. All maxillary (Fig 5A) and mandibular (Fig 5B) teeth revealed severe recession of the alveolar bone; several

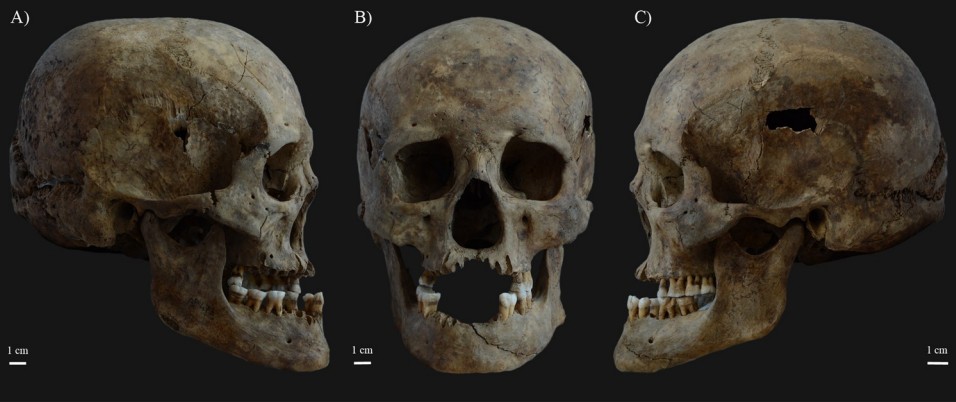

**Fig 3.** A) Right lateral, B) anterior, and C) left lateral view of the skull of KK61, with severe bony changes indicative of leprosy in the rhinomaxillary region of the face.

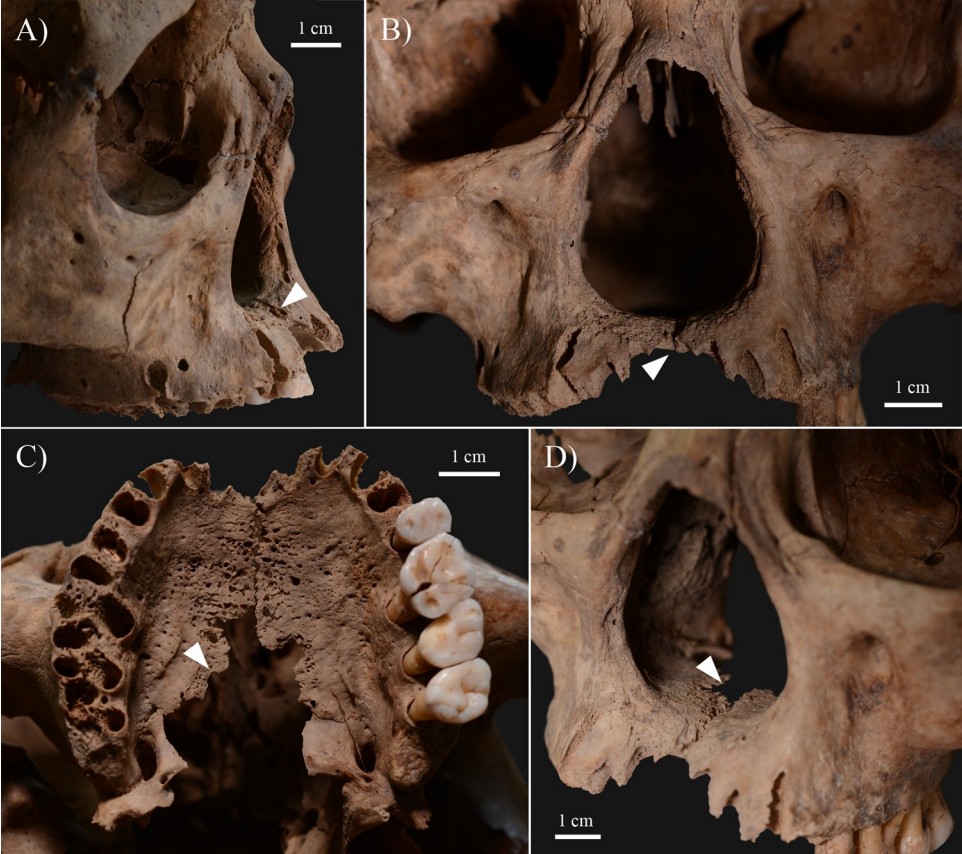

**Fig 4. Rhinomaxillary lesions in the skull of KK61.** A) Resorption of the anterior nasal spine (white arrow) and the left inferior nasal concha, and widening and rounding of the inferior half of the pyriform aperture (left side); B) Resorption of the anterior nasal spine, the bony nasal septum, the inferior nasal conchae, and the maxillary alveolar process (at the prosthion–white arrow), and widening and rounding of the inferior half of the pyriform aperture; C) Resorption of the maxillary alveolar process (at the prosthion) and pitting, erosion, and perforation (white arrow) on the oral surface of the maxillary palatine process; and D) Resorption of the anterior nasal spine, the bony nasal septum, the right inferior nasal concha, and the maxillary alveolar process (at the prosthion), widening and rounding of the inferior half of the pyriform aperture, and pitting, erosion, and perforation (white arrow) on the nasal surface of the maxillary palatine process.

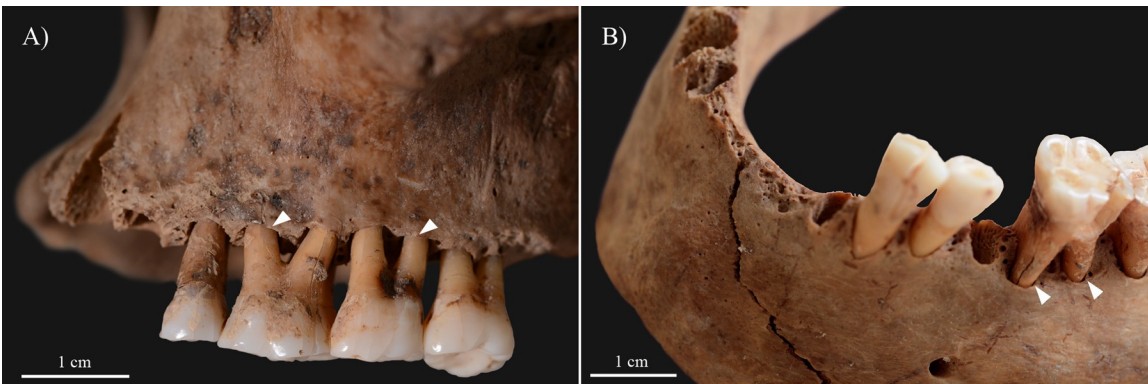

**Fig 5.** Alveolar bone recession (white arrows) of the A) maxillary and B) mandibular teeth of KK61, with *ante-mortem* loss of the mandibular central incisors.

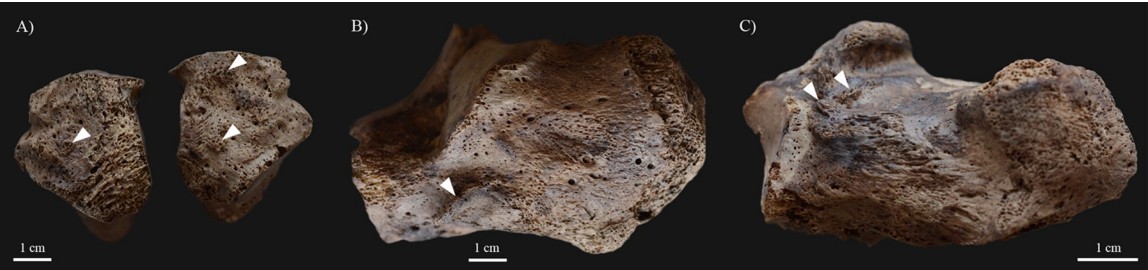

**Fig 6. Bony changes indicative of motor peripheral neuropathy in the feet of KK61.** A) Exostoses on the dorsal surface of the left and right cuboid bones (white arrows); B) Bony ridges on the lateral surface of the left calcaneus (white arrow); and C) Exostoses on the medial surface of the right calcaneus (white arrows).

of them also exhibited signs of dental calculus and/or caries (S1 Table). The mandibular central incisors were lost *ante-mortem* with consequent filling up of their alveoli with new bone (Fig 5B).

## Postcranial bony changes

In the postcranial skeleton of **KK61**, only the tarsal bones and metatarsals of the feet (Figs 6 and 7), and the long tubular bones of the lower legs (Fig 8) displayed alterations that are indicative of HD. Although the *post-mortem* missing and damages precluded the definitive observation of some of the tarsal bones, the left and right cuboid bones (Fig 6A), and the right medial cuneiform bone exhibited exostoses with a height of few millimetres, at the attachment sites of the dorsal tarsal ligaments. The left and right calcanei revealed bony ridges on their lateral surfaces (at the attachment sites of the calcaneofibular, and the lateral and interosseous talocalcaneal ligaments) (Fig 6B) and exostoses with a height of few millimetres on their medial surfaces (at the attachment sites of the spring ligament) (Fig 6C). The dorsal and plantar surfaces of the proximal end of the left 3$^{rd}$, 4$^{th}$, and 5$^{th}$ (Fig 7A and 7B), and the right 3$^{rd}$ and 4$^{th}$ metatarsals displayed septic bony changes in the form of surface pitting and subperiosteal new bone formations that were most pronounced on the left 5$^{th}$ metatarsal. In addition, there were small sinuses on the dorsal surface of the left 5$^{th}$ (Fig 7A), the lateral surface of the right 3$^{rd}$, and the medial surface of the right 4$^{th}$ metatarsals. In the right 2$^{nd}$ metatarsal (Fig 7C), slight ballooning of the diaphysis was observed. The distal end of the 1$^{st}$ right metatarsal (Fig 7D and 7E) was almost completely destroyed, whereas the plantar surface of its proximal end (Fig 7F) presented remodelling with surface pitting and subperiosteal new bone formations. Similar alterations were detected on the plantar surface of the right medial cuneiform bone (Fig 7F and 7G). Both tibiae (Fig 8A) and fibulae (Fig 8B) revealed slight surface pitting and longitudinally striated subperiosteal new bone formations with a more or less organised, lamellar-like macroscopic appearance. The aforementioned lesions were most pronounced along the medial surface of the tibial shafts, and the medial and posterior surfaces of the fibular shafts. Furthermore, on the distal part of both tibiae (Fig 8C) and fibulae (Fig 8D), there were some exostoses with a height of few millimetres at the attachment sites of the crural interosseous membrane. Similar exostoses were observed on the *linea aspera* of both femora (Fig 8E).

## Discussion and conclusions

Local invasion of the bones in the rhinomaxillary region of the face by leprosy bacilli, either via direct extension of the infection from the adjacent soft tissues (e.g., skin or oronasal mucosa) or via haematogenous spread of the pathogens, leads to the formation of rhinomaxillary bony

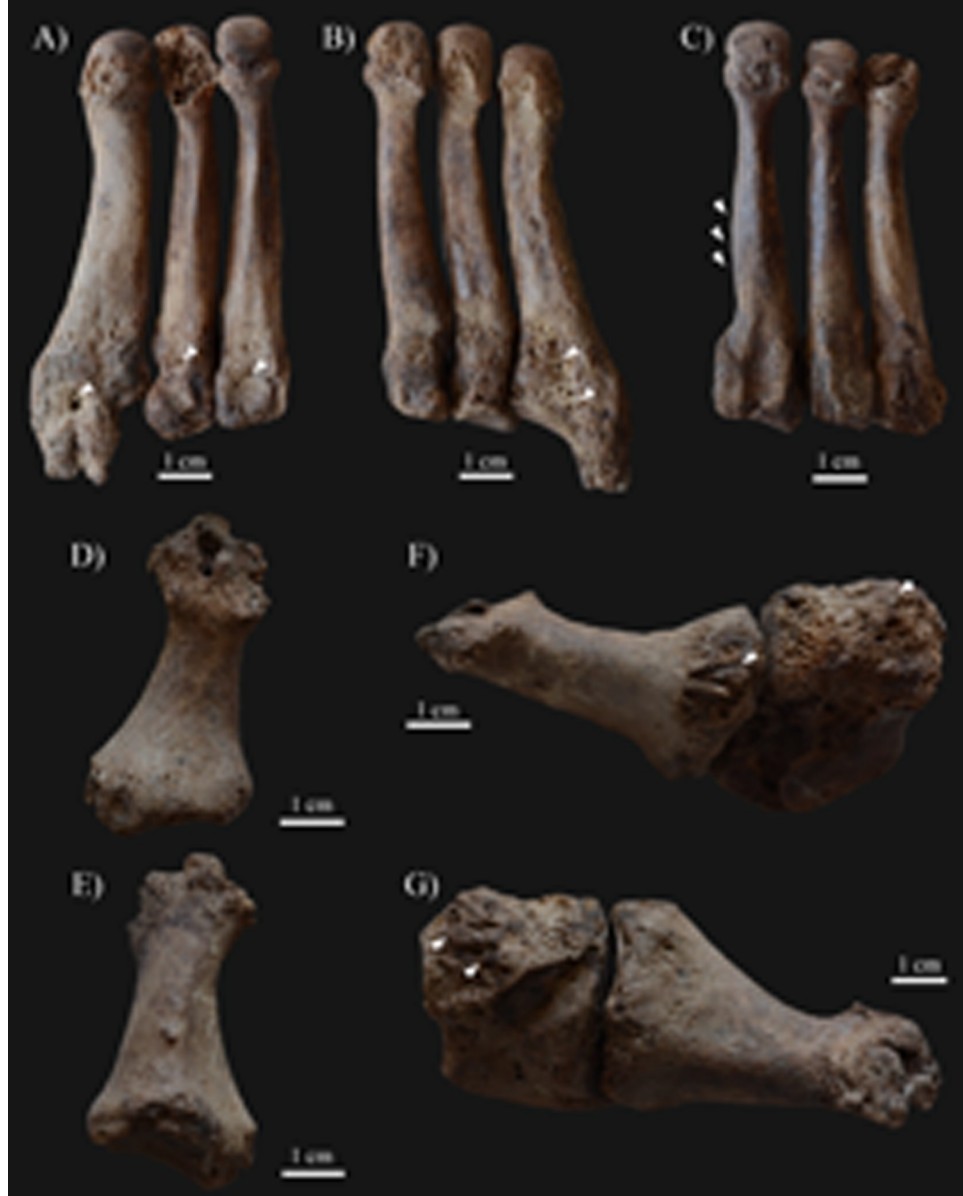

**Fig 7. Septic bony changes indicative of sensory peripheral neuropathy in the feet of KK61.** Surface pitting and subperiosteal new bone formations on the A) dorsal and B) plantar surfaces of the proximal end of the left 3rd, 4th, and 5th metatarsals (white arrows), with a small sinus on the 5th metatarsal; C) Slight ballooning of the diaphysis of the right 2nd metatarsal (white arrows); Almost complete destruction of the distal end of the right 1st metatarsal–D) plantar and E) dorsal surfaces; and Remodelling of the proximal end of the right 1st metatarsal and the right medial cuneiform bone with surface pitting and subperiosteal new bone formations (white arrows)–F) right lateral and G) left lateral view.

changes [8–10]. These alterations are collectively referred to as '*facies leprosa*' or rhinomaxillary syndrome (RMS) [32, 33, 36]. RMS is a composite mixture of absorptive, erosive, and proliferative lesions that involve the anterior nasal spine, the pyriform aperture (especially its lateral and inferior margins), the alveolar and palatine processes of the maxilla, and the intranasal bony structures, such as the bony nasal septum and the inferior nasal conchae [9, 10, 17,

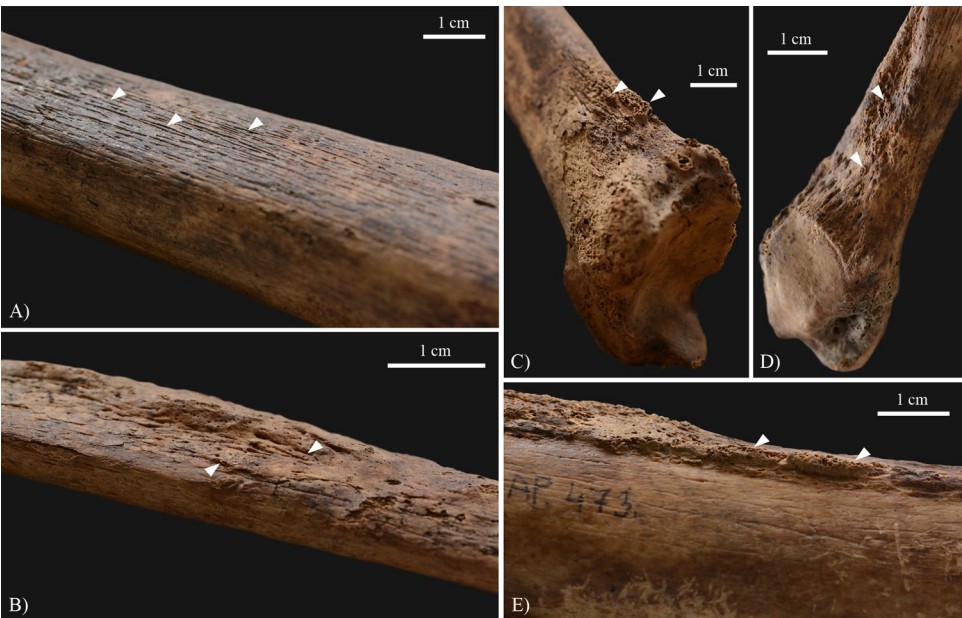

**Fig 8.** Slight surface pitting and longitudinally striated subperiosteal new bone formations (white arrows) on the shaft of the right A) tibia (medial surface) and B) fibula (posterior surface) of KK61. Exostoses (white arrows) on the distal part of the right C) tibia (lateral surface), D) fibula (medial surface), and E) femur (posterior surface) of KK61.

36]. RMS is pathognomonic for near-lepromatous or lepromatous leprosy [8, 17, 36]. The rhinomaxillary bony changes observed in **KK61** (Figs 3 and 4) are compatible with RMS.

In the middle-aged male, the mucous membrane lining the nasal cavity could have been the initial site of the leprous infection. Following invasion of the nasal mucosa by leprosy bacilli, atrophic rhinitis, i.e., progressive thinning and hardening of the nasal mucosa, could have developed. This condition is characterised by symptoms of highly bacilliferous, muco-purulent or exudative, foul-smelling nasal discharges, dry, thick nasal crusts, recurrent epistaxis, and obstruction of the nasal airways [36, 46, 47]. In later stages of the disease, not only the nasal mucosa but the underlying cartilaginous and bony nasal structures (e.g., nasal septum, inferior nasal conchae, and nasal surface of the maxillary palatine process) could have been affected by the pathological process, with their subsequent destruction (Fig 4A, 4B and 4D). Loss of the cartilaginous and bony support in the nose of **KK61** could have resulted in the formation of saddle-nose deformity, i.e., abnormal concavity of the nasal dorsum [3, 46, 48]. It is characteristic of advanced-stage lepromatous leprosy and can provoke not only aesthetic repercussions (e.g., facial disfigurement) but also functional implications (e.g., impaired nasal breathing) [36, 44, 46, 48].

Because of the persistent blockage of his nasal airways, the middle-aged male could have experienced difficulties in breathing through his nose; and thus, he could have developed an oral breathing habit that is a frequent accompaniment of lepromatous leprosy [44, 49, 50]. As oral breathing can contribute to changes in the intraoral humidification, pH, and oxygen levels, it can have a negative impact on oral health [51]. For instance, chronic oral breathing can increase the risk of developing periodontitis, dental calculus or caries [51, 52]. Therefore, HD and consequent breathing through the mouth cannot be excluded from being at least partially responsible for the alveolar bone recession, dental calculus and caries, and *ante-mortem* tooth loss detected in **KK61** (Fig 5). Based on the severity of the observed alveolar bone loss in the

mandible and maxillae, it can also be presumed that the middle-aged male had advanced-stage periodontitis; and thus, he would have suffered from pronounced halitosis [53, 54].

During oral breathing, the cooling effect of the inspired air and evaporation lowers the surface temperature of certain areas in the oral cavity, such as the anterior part of the maxilla, and the soft and hard palates [49, 50, 55, 56]. Because of the lower surface temperature, the mucous membrane of these oral sites becomes more prone to seeding by leprosy bacilli and consequently, to developing leprous lesions [49, 50, 56]. Later, not only the oral mucosa but the underlying bone, such as the premaxilla and the oral surface of the maxillary palatine process, can become affected by the infection, with subsequent formation of oral bony changes (e.g., perforation of the hard palate) [36, 49, 50]. In the middle-aged male, leprous involvement of both the nasal and oral surfaces of the maxillary palatine process could have terminated in perforation of the hard palate (Fig 4C and 4D). It could have been accompanied by the development of an oronasal fistula, i.e., an abnormal communication between the nasal and oral cavities, and associated functional difficulties, such as speech impairments, and nasal regurgitation of food and liquids [44, 50, 57].

Invasion of the peripheral nerves by leprosy bacilli and consequent inflammatory changes in the affected nerves can lead to loss of sensory, motor, and/or autonomic nerve functions [8, 34, 58]. The large, superficial peripheral nerves, such as the ulnar and posterior tibial nerves, are characteristically affected in HD [35, 59]. Leprous neuropathy in one or more peripheral nerves can result in the formation of bony changes in the small bones of the hands and/or feet (e.g., palmar grooving at the distal end of the proximal hand phalanges, palmar bevelling at the proximal end of the middle hand phalanges, exostoses on the dorsal surface of the tarsal bones, concentric diaphyseal atrophy and/or achro-osteolysis of the metacarpals, metatarsals, and hand and/or foot phalanges, and septic bony and articular lesions) and the long tubular bones of the arms and/or legs (e.g., surface pitting and subperiosteal new bone formations) [8, 10, 11, 34, 35, 37–39]. Although these lesions are highly characteristic of leprosy, they cannot be considered as pathognomonic features of the disease [38]. In the postcranial skeleton of **KK61**, several alterations indicative of leprous neuropathy in the left and right posterior tibial nerves (Figs 6–8) have been observed. Bilateral, symmetrical, distal polyneuropathy of the peripheral nerves is typical of lepromatous leprosy [3, 60].

In **KK61**, the presence of peripheral motor neuropathy can be presumed in both feet, as the exostoses on the dorsal surface of several tarsal bones (Fig 6) serve as direct evidence for leprous involvement and consequent motor dysfunction of the left and right posterior tibial nerves [35]. Due to motor impairment in the posterior tibial nerves, the muscles, which are responsible for the maintenance of the longitudinal arch integrity in the foot, could have become paralysed [35]. This could have led to collapse of the longitudinal arch with subsequent development of 'flat-foot' deformity in both of the middle-aged male's feet [35]. Due to the changed and abnormal mechanical stress between the tarsal bones in his deformed feet, there could have been dynamic, progressive, plantar displacement of the navicular bones [35]. This could have caused strain on the dorsal tarsal ligaments with secondary formation of exostoses at their attachment sites on the dorsal surface of the tarsal bones [11, 35]. In **KK61**, the development of flat-foot deformity could have been accompanied by foot disfigurement and difficulties in standing and walking, and thereby in conducting domestic and occupational physical activities [61].

Based on the presence of pyogenic septic lesions on the small bones of the feet (Fig 7) and the long tubular bones of the lower legs of **KK61** (Fig 8), not only the motor but also the sensory component of the posterior tibial nerve could have been affected by HD in the middle-aged male's both lower limbs [38]. Early in the course of leprosy, involvement of the posterior tibial nerve, that supplies sensation to the plantar foot, could have resulted in loss of cutaneous

sensation and consequent formation of plantar ulcers in both feet of **KK61** [62]. This is because his insensitive feet could have become more prone to repeated, unperceived, superficial traumata to the soft tissues (e.g., burns or cuts) and secondary pyogenic sepsis [8, 38, 39, 63]. Nevertheless, even the customary use of the middle-aged male's feet in standing and walking could have led to their plantar ulceration, especially in their weight-bearing areas [38, 64]. The site of trauma and ulceration is influenced by the developed foot deformity type, in correlation with the altered plantar weight distribution of the body weight [11, 38]. In case of flatfoot deformity, which can be presumed in **KK61**, plantar trauma and ulceration tend to occur in the mid-foot, particularly in the region of the tarsometatarsal joints [38].

In more advanced stages of HD, with the loss of not only the cutaneous but also the deep tissue sensation, the pyogenic sepsis could have extended from the superficial soft tissues to the deep soft tissues, bones, and joint cavities of the middle-aged male's feet [8, 38–39]. This could have given rise to the formation of the pyogenic septic lesions detected on several tarsal bones and metatarsals of **KK61** (e.g., surface pitting, subperiosteal new bone formations, and partial bone absorption) (Fig 7) [8, 38]. These alterations serve as direct evidence for the *in vivo* presence of plantar ulceration of the middle-aged male's both feet [38]. Their localisation and severity pattern in the tarsal bones and metatarsals of **KK61** suggest that the *in vivo* plantar ulcers could have particularly been found in the mid-foot, strengthening the evidence that flatfoot deformity was present in the middle-aged male's both feet [35, 38].

Signs of periostitis on the left and right tibiae and fibulae, in the form of slight surface pitting and longitudinally striated subperiosteal new bone formations (Fig 8A–8D), imply that the pyogenic infection could have ascended from the plantar ulcers of his feet to his lower legs [38]. It is important to note that in cases with leprosy, signs of periostitis in the lower leg bones appear to develop more frequently when accompanied by mid-foot plantar ulceration [64], which can be presumed in **KK61**. The more organised, lamellar-like macromorphological appearance of the detected subperiosteal new bone formations suggests that the infection occurred and healed before the middle-aged male's death [39, 40]. Besides the tibiae and fibulae, the crural interosseous membrane could have become involved by the pyogenic infection in both lower legs of **KK61** [39]. Inflammation and subsequent ossification of this ligamentous structure could have led to the formation of exostoses at its attachment sites on the adjoining tibia and fibula (Fig 8C and 8D) [39].

In the middle-aged male, the foot bones do not provide any skeletal evidence for the development of autonomic peripheral neuropathy. Nonetheless, in HD, it is an invariable accompaniment of impaired cutaneous and/or deep tissue sensation [38]. Therefore, based on the detected bony changes indicative of combined sensory and motor dysfunction of the left and right posterior tibial nerves, the presence of autonomic peripheral neuropathy can be presumed in both lower limbs of **KK61**. The autonomic dysfunction in his feet could have been associated with anhidrosis, with his anhidrotic, and thereby xerotic and hyperkeratotic plantar skin becoming more prone to ulceration and secondary pyogenic sepsis [59, 65, 66]. Furthermore, peripheral autonomic impairment could have led to loss of vascular tone and subsequent stasis of the capillary blood flow in the middle-aged male's both feet, with consequent delayed ulcer healing [65, 66]. In patients with HD, non-healing plantar ulcers are a common cause of disability, and thereby of reduced quality of life [67, 68].

In summary, although several differential diagnoses need to be considered for the single alterations detected in the skeleton of **KK61**, based on their nature, association, and distribution pattern, they are most likely compatible with near-lepromatous or lepromatous leprosy [69]. The severity and extent of the observed bony changes imply that the middle-aged male suffered from HD for a long time prior to his death. The disease has affected not only the rhinomaxillary region of his face but his lower limbs, with severe deformation and disfigurement

of the involved anatomical areas (saddle-nose and flat-foot deformity, respectively). Consequently, he would have experienced disability in performing the basic activities of daily living, such as eating, drinking, standing or walking; and thus, he would have required regular and substantial care from others to survive.

Despite his very visible disease and associated debility, that clearly marked the middle-aged male as afflicted with leprosy, he has not been segregated but buried within the cemetery boundaries, among others from his community (Fig 1C). His grave has conformed to the mortuary practices characteristic of the Kiskundorozsma–Kettőshatár I cemetery (e.g., burial orientation, position of the body in the grave, and type and quantity of accompanying grave goods) (Fig 2A). Based on the above, there seems to have been no distinction, leper from non-leper, in death in the late Avar Age community of Kiskundorozsma–Kettőshatár I. These findings are in accordance with the results of previous studies on other Avar Age cemeteries from the present-day territory of Hungary, indicating that prior to the 13th century CE, lepers have not been segregated from the healthy population, at least in death [70]. It should be noted that distinction or segregation in life do not preclude normative treatment in death. Nevertheless, the long-lasting survival of **KK61** with HD indicates that he would not have been abandoned but cared for by others.

The detailed contextual analysis of **KK61** illuminates both the biological and social consequences of living and dying with leprosy in the late Avar Age community of Kiskundorozsma–Kettőshatár I. In the future, mycobacterial aDNA analyses are planned to be performed on **KK61.** On the one hand, these investigations could yield an independent and clear evidence of the infection. On the other hand, they could provide us with invaluable information about the origins and geographical distribution of leprosy bacilli, and the migration routes of their human host over time.

## Supporting information

**S1 Table. Dental status of KK61, and location, direction, and grade of caries and calculus on his affected teeth.**
(PDF)

## Author Contributions

**Conceptualization:** Olga Spekker.

**Data curation:** Olga Spekker.

**Funding acquisition:** Olga Spekker, Tivadar Vida, György Pálfi.

**Investigation:** Olga Spekker, Balázs Tihanyi, Csaba Szalontai, Antónia Marcsik, Erika Molnár.

**Methodology:** Olga Spekker.

**Project administration:** Olga Spekker.

**Resources:** György Pálfi.

**Supervision:** Tivadar Vida, György Pálfi.

**Visualization:** Olga Spekker, Luca Kis, Csaba Szalontai.

**Writing – original draft:** Olga Spekker.

**Writing – review & editing:** Olga Spekker.

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
