## [Decision Letter · Decision Letter 0]

13 Jan 2022

PONE-D-21-32338Life and death of a leprosy sufferer from the 8th-century-CE cemetery of Kiskundorozsma–Kettőshatár I (Duna-Tisza Interfluve, Hungary) – Biological and social consequences of having Hansen’s disease in a late Avar Age population from HungaryPLOS ONE

Dear Dr. Spekker,

Thank you for submitting your manuscript to PLOS ONE. After careful consideration, we feel that it has merit but does not fully meet PLOS ONE’s publication criteria as it currently stands. Therefore, we invite you to submit a revised version of the manuscript that addresses the points raised during the review process.

Both reviewers agree (and I tend to agree with them) that this is a very well written and interesting study that should be of interest to the professionals as well as to a wider audience. The reviewers did not find any major issues with the manuscript, but they did list a number of smaller issues that have to be dealt in a proper manner (I won't mentioned these here). Please, look carefully at their suggestions and revise the text accordingly.

We look forward to receiving your revised manuscript.

Kind regards,

Mario Novak

Academic Editor

PLOS ONE

Journal Requirements:

Reviewers' comments:

Reviewer's Responses to Questions

**Comments to the Author**

1. Is the manuscript technically sound, and do the data support the conclusions?

Reviewer #1: Yes

Reviewer #2: Yes

2. Has the statistical analysis been performed appropriately and rigorously? 

Reviewer #1: N/A

Reviewer #2: N/A

3. Have the authors made all data underlying the findings in their manuscript fully available?

Reviewer #1: Yes

Reviewer #2: Yes

4. Is the manuscript presented in an intelligible fashion and written in standard English?

Reviewer #1: Yes

Reviewer #2: Yes

5. Review Comments to the Author

Reviewer #1: I’ve read with interest this paper about a probable case of leprosy from an Avar cemetery of Hungary. The case is very well described and the article is well written. Some minor points need revisions.

Page 6, lines 134: please specify to which period does the middle Avar Aga correspond.

Page 7, line 163: does the last phase of the cemetery correspond to 9th century?

Page 10 line 214: it could be interesting to specify which teeth were affected by caries and of which grade. From figure 4 it seems that the majority of maxillary teeth were lost post-mortem.

Post-cranial bone changes: any lesions was observed in the hand bones? Please discuss this issue.

Page 18, line 393: it is not clear which is the possible skeletal evidence for the development of autonomic peripheral neuropathy.

In the discussion it should be interesting to refer about the other few cases of leprosy found in the same region (as stated for example in the abstract and at page 2). Were also these individuals affected by leprosy found in the Hungarian Duna-Tisza Interfluve buried within the cemetery boundaries among the others from his community? This could indicate between the Avars a diffuse practice of inclusion of severely sick individuals, who were not segregated, as historically attested in Medieval Europe.

As the work is within a Synergy project (as stated in the financial disclosure) based on molecular analysis, it is not clear why molecular analysis was not carried out in order to confirm the diagnosis. Please justify this issue.

Reviewer #2: The paper Life and death of a leprosy sufferer from the 8th-century-CE cemetery of Kiskundorozsma–Kettőshatár I (Duna-Tisza Interfluve, Hungary) – Biological and social consequences of having Hansen’s disease in a late Avar Age population from Hungary is worth to be published in PLOS ONE with minor revisions. The overall impression is that this kind of study using all known and acknowledged diagnostic criteria for leprosy is very helpful especially when the skeleton is very well preserved like was the case with this one. Special weight and importance of the work is given by a series of clinical studies on the basis of which it was possible to reconstruct what was happening to a male infected with leprosy. Since there is a scarce of published historic cases with leprosy from the Avar Age of the Hungarian Duna-Tisza Interfluve the importance of publishing this study is even greater.

Minor changes should be made:

In the Title, in Abstract on page 2, line 24 and in Aims on page 6, lines 138-139 I don't think that dashes in 8th-century-CE are needed.

In Aims on page 6, line 138 a middle-aged male should be replaced with a skeleton. I think that in this part of the manuscript age and sex of the skeleton still shouldn't be revealed. Also on page 7, line 147.

In part Burial context, on page 7, line 154 for a middle-aged (c.45-50 years old) male should be put the methodology or criteria on the basis of which the sex and age of the skeleton were determined.

Although according to diagnostic criteria for leprosy and given photographs of the skeleton it is clear that this is the case of lepromatous leprosy, but maybe it should be mentioned somewhere in the text if molecular genetic analysis was performed to confirm the presence of the disease.

Is this the only case of leprosy from this site? It seems unlikelly considering this is an infectious dissease and this is one of the largest Avar age cemeteries in the region with almost 300 skeletons.

6. PLOS authors have the option to publish the peer review history of their article (what does this mean?). If published, this will include your full peer review and any attached files.

Reviewer #1: No

Reviewer #2: **Yes: **Željka Bedić

---

## [Author Response · Author response to Decision Letter 0]

17 Jan 2022

Dear Dr Mario Novak,

I am very thankful for the reviewers’ insightful and constructive comments regarding our manuscript entitled “Life and death of a leprosy sufferer from the 8th-century-CE cemetery of Kiskundorozsma–Kettőshatár I (Duna-Tisza Interfluve, Hungary) – Biological and social consequences of having Hansen’s disease in a late Avar Age population from Hungary” that was submitted to the PLOS ONE (manuscript ID: PONE-D-21-32338). I am sure that the reviewers helped us to improve the quality of our manuscript. The main text has been modified following the reviewers’ suggestions. Furthermore, a supplementary table (S1 Table) has been included. The revised files have been uploaded to the submission site of the PLOS ONE.

Responses to the suggestions:

1) Reviewer 2 noted that “In the Title, in Abstract on page 2, line 24 and in Aims on page 6, lines 138–139 I don’t think that dashes in 8th-century-CE are needed.”. In the above-mentioned sentences, we used the hyphenated form “8th-century-CE” to join the words into one, as we used “8th-century-CE” as a qualificative before “cemetery” (it is similar to “1-year-old” child or “middle-aged” male, etc.). If Reviewer 2 agrees with us, we would like to keep the hyphenated form in the aforementioned sentences.

2) Reviewer 2 commented that “In Aims on page 6, line 138 a middle-aged male should be replaced with a skeleton. I think that in this part of the manuscript age and sex of the skeleton still shouldn’t be revealed. Also on page 7, line 147.” To not reveal the age and sex information of the examined individual in the Aims section, this part of the manuscript was slightly rephrased:

“The aim of our paper is to demonstrate an individual (KK61) from the 8th-century-CE cemetery of Kiskundorozsma–Kettőshatár I (Duna-Tisza Interfluve, Hungary), who appears to represent the lepromatous form of HD. The case of KK61 has been only briefly summarised in a review by Marcsik and her colleagues [21] but the detailed macromorphological description of the bony changes observed in the skeleton has not yet been provided. To reconstruct the type of leprosy and the biological consequences of the disease progression in KK61, the detected leprous lesions were linked with palaeopathological and modern medical information. Furthermore, to reconstruct the social consequences of being afflicted with HD in the late Avar Age community of Kiskundorozsma–Kettőshatár I, conceptualisation of the examined individual’s treatment in death was conducted. KK61 is one of the few published historic cases with leprosy from the Avar Age of the Hungarian Duna-Tisza Interfluve. The case of KK61 gives us a unique insight into the biological consequences of living with HD and illustrates the social attitude toward leprosy sufferers in early mediaeval Hungary.”

3) Reviewer 1 asked us to “specify to which period does the middle Avar Aga correspond” in page 6, line 134. Reviewer 1 also asked “does the last phase of the cemetery correspond to 9th century?” in page 7, line 163. Following Reviewer 1’s comments, the relevant sentences were supplemented with the requested information:

Page 6, line 134: “Based on the associated grave goods, the cemetery of Kiskundorozsma–Kettőshatár I may have been in use from the end of the middle Avar Age (last third of the 7th century CE) until the turn from the 8th to the 9th century CE [24-26].”

Page 7, line 163: “Burial 61 can be dated to the last phase (late 8th–early 9th century CE) of the Kiskundorozsma–Kettőshatár I cemetery [25].”

4) Reviewer 2 suggested that “In part Burial context, on page 7, line 154 for a middle-aged (c.45-50 years old) male should be put the methodology or criteria on the basis of which the sex and age of the skeleton were determined.”. Following Reviewer 2’s suggestion, the methods used to estimate the age-at-death and to determine the sex of KK61 were included to the Methods section. In the Burial context and Methods sections, the relevant sentences were amended accordingly: 

Burial context: “The skeleton of KK61, a middle-aged (c. 45–50 years old) male, was unearthed from burial 61 of the Kiskundorozsma–Kettőshatár I cemetery (Fig 1C). …”

Methods: “The relatively complete and well-preserved skeleton of KK61 (Fig 2B) was subject to a detailed macromorphological analysis, focusing on the detection of pathological bony changes probably related to leprosy. Prior to the palaeopathological examination of KK61, age-at-death was estimated and sex was determined applying standard macromorphological methods of bioarchaeology [28-31]. Based on the results of these investigations, the skeleton belonged to a middle-aged (c. 45–50 years old) male. During the palaeopathological evaluation, all skeletal remains of KK61 were macroscopically examined with the naked eye. …”. 

Because of the inclusion of references regarding the used age-at-death estimation and sex determination methods, the subsequent references were re-numbered throughout the main text. Furthermore, the above references were included to the reference list, as well.

5) Reviewer 1 noted that in page 10, line 214, “it could be interesting to specify which teeth were affected by caries and of which grade. From figure 4 it seems that the majority of maxillary teeth were lost post-mortem.” Following Reviewer 1’s suggestion, a supplementary table (S1 Table) containing information about the dental status of KK61 and the grade of caries on his affected teeth was included.

6) Reviewer 1 asked us if “any lesions was observed in the hand bones? Please discuss this issue”. It seems that our phrasing was not clear enough, when we stated that “In the postcranial skeleton of KK61, the tarsal bones and metatarsals of the feet (Figs 6 and 7), and the long tubular bones of the lower legs (Fig 8) displayed alterations that are indicative of HD.” in page 11, lines 234–236 in the original version of our manuscript. We are really sorry for that. 

To avoid confusion, the text was amended: “In the postcranial skeleton of KK61, only the tarsal bones and metatarsals of the feet (Figs 6 and 7), and the long tubular bones of the lower legs (Fig 8) displayed alterations that are indicative of HD.” We hope that this rephrasing will properly indicate that the hand bones of KK61 did not reveal any leprosy-related lesions.

7) Reviewer 1 commented that in page 18, line 393 “it is not clear which is the possible skeletal evidence for the development of autonomic peripheral neuropathy”. Autonomic peripheral neuropathy can manifest itself in the skeleton as concentric diaphyseal atrophy of the metacarpals, metatarsals, and hand and/or foot phalanges that can be accompanied by achro-osteolysis (reference 33 in the original version of our manuscript). As we discussed in page 18, lines 392–397, although no skeletal sign of autonomic peripheral neuropathy could be observed in KK61, its presence can be presumed in both of his lower limbs. It is because of the presence of bony changes related to sensory and motor peripheral neuropathy in both posterior tibial nerves of KK61. (In HD, autonomic peripheral neuropathy invariably accompanies sensory peripheral neuropathy, which precedes motor peripheral neuropathy (reference 34 in the original version of our manuscript).)

“In the middle-aged male, the foot bones do not provide any skeletal evidence for the development of autonomic peripheral neuropathy. Nonetheless, in HD, it is an invariable accompaniment of impaired cutaneous and/or deep tissue sensation [34]. Therefore, based on the detected bony changes indicative of combined sensory and motor dysfunction of the left and right posterior tibial nerves, the presence of autonomic peripheral neuropathy can be presumed in both lower limbs of KK61.”

8) Reviewer 1 mentioned that “In the discussion it should be interesting to refer about the other few cases of leprosy found in the same region (as stated for example in the abstract and at page 2). Were also these individuals affected by leprosy found in the Hungarian Duna-Tisza Interfluve buried within the cemetery boundaries among the others from his community? This could indicate between the Avars a diffuse practice of inclusion of severely sick individuals, who were not segregated, as historically attested in Medieval Europe.”

Until now, only a few anthropological studies discussing leprosy cases from the Avar Age of the Hungarian Duna-Tisza Interfluve have been published (e.g., references 21 and 22 in the original version of our manuscript), and in these papers, the aforementioned distinction/segregation aspect has not been developed. Nevertheless, it was briefly discussed in a poster presentation (reference 70 in the revised version of our manuscript) that reviewed “the evidence of skeletal leprosy from ancient times in the territory of present-day Hungary based on previously published and newly described cases”. In this study, it was stated that: “After the 13th–14th centuries leprosy is not present in skeletal remains in community cemeteries. This fact could be explained by the establishment of leprosaria and/or the decline of this infection. From this time leprous individuals were isolated from the non-infected population. Contrary to this, according to the analysis of cemetery plans, no sign of segregation of infected people have been found prior to the 13th century AD.”. 

Following Reviewer 1’s advice, the above-mentioned information was included to the Discussion and Conclusions section in the revised version of our manuscript: “… Based on the above, there seems to have been no distinction, leper from non-leper, in death in the late Avar Age community of Kiskundorozsma–Kettőshatár I. These findings are in accordance with the results of previous studies on other Avar Age cemeteries from the present-day territory of Hungary, indicating that prior to the 13th century CE, lepers have not been segregated from the healthy population, at least in death [70]. It should be noted that distinction or segregation in life do not preclude normative treatment in death. …”

9) Reviewer 1 noted that “As the work is within a Synergy project (as stated in the financial disclosure) based on molecular analysis, it is not clear why molecular analysis was not carried out in order to confirm the diagnosis. Please justify this issue.”. Reviewer 2 commented that “Although according to diagnostic criteria for leprosy and given photographs of the skeleton it is clear that this is the case of lepromatous leprosy, but maybe it should be mentioned somewhere in the text if molecular genetic analysis was performed to confirm the presence of the disease.”.

We agree with the reviewers that positive results of a palaeomicrobiological investigation (e.g., mycobacterial aDNA. lipid biomarker or peptide analyses) could strengthen the macromorphology-based diagnosis of leprosy in KK61. Nevertheless, we think that in the case of KK61, the diagnosis is supported enough based on the detected bony changes, especially their nature, co-occurrence, and distribution in the skeleton – Reviewer 2 also mentioned in her review that “according to diagnostic criteria for leprosy and given photographs of the skeleton it is clear that this is the case of lepromatous leprosy”. On the other hand, the chance for successfully detecting Mycobacterium leprae DNA, lipid biomarkers or peptides in historic leprosy cases depends on a number of factors (e.g., state of preservation of the extant bone remains). As an adage in aDNA says: “absence of evidence is not evidence of absence”. Thus, even if a palaeomicrobiological analysis would give negative results, it would not mean that KK61 did not suffer from leprosy. In our opinion, the above-mentioned palaeomicrobiological analyses are very useful to complement the findings of traditional macromorphology-based investigations, but should not be considered as exclusive and indispensable tools to confirm a diagnosis.

Reviewer 1 is right that an ERC Synergy project („HistoGenes” 856453 ERC-2019-SyG) financially supports our investigations regarding the Kiskundorozsma–Kettőshatár I cemetery, and aDNA analyses are carried out in a number of skeletons (c. 6,000). Nonetheless, these aDNA analyses primarily focus on the human genome. Of course, some pathogen aDNA analyses (c. 500) are also planned. KK61 is on the list that contains skeletons, which should be examined from this aspect, but these investigations have not yet been performed on KK61. If in the future, M. leprae aDNA could be detected in the tooth sample from KK61, it would obviously strengthen the macromorphology-based diagnosis of leprosy, but more importantly, its (sub)genotyping could provide us with invaluable information about the origins and geographical distribution of leprosy bacilli, and the migration routes of their human host over time. Following Reviewer 2’s suggestion, at the very end of the revised version of our manuscript, it is mentioned that in the future, mycobacterial aDNA analyses are planned: “In the future, mycobacterial aDNA analyses are planned to be performed on KK61. On the one hand, these investigations could yield an independent and clear evidence of the infection. On the other hand, they could provide us with invaluable information about the origins and geographical distribution of leprosy bacilli, and the migration routes of their human host over time.”

In summary, we think that the anthropological data discussed in our manuscript, on their own, are worth to be published. Of course, if aDNA and other palaeomicrobiological examinations could be performed on KK61 in the future, they would further extend our knowledge regarding this case, as well as the past of leprosy in the Hungarian Duna-Tisza Interfluve. I hope the reviewers will agree with us.

10) Reviewer 2 asked us if “Is this the only case of leprosy from this site? It seems unlikely considering this is an infectious disease and this is one of the largest Avar age cemeteries in the region with almost 300 skeletons.”.

Based on the findings of previous macromorphological investigations (reference 21 in the original version of our manuscript), from the Avar Age cemetery of Kiskundorozsma–Kettőshatár I, only another middle-aged (c. 40–49 years old) male (KK245) exhibited bony changes (only in the rhinomaxillary region) that could indicate the diagnosis of leprosy. However, the detected skeletal lesions were not that conclusive; and thereby, the biological consequences and the form of the disease could not be so convincingly reconstructed, to include this individual into our paper. (It should be noted that KK245 was also buried within the cemetery boundaries.) Interestingly, from the geographically very closely located Avar Age cemeteries of Kiskundorozsma–Kettőshatár II (43 graves) and Kiskundorozsma–Daruhalom-dűlő II (93 graves), one and seven cases with leprosy have been described by Marcsik and her co-workers (reference 21 in the original version of our manuscript), respectively. We agree with Reviewer 2 that we should expect more cases with leprosy from the Kiskundorozsma–Kettőshatár I cemetery, especially if we consider that in the much smaller Kiskundorozsma–Daruhalom-dűlő II cemetery, a high number of cases have been identified during the previous investigations (reference 21 in the original version of our manuscript). Therefore, re-evaluation of the three Kiskundorozsma cemeteries, focusing on leprosy-related bony changes, is in progress. In the first phase of the project, we re-evaluated the cases briefly described by Marcsik and her colleagues in 2007 (one of them was KK61) – provided the detailed description of the observed leprosy-related skeletal lesions and if possible, reconstructed the form of the disease, as well as its biological and/or social consequences. In the second phase of the project, we will re-evaluate all the individuals from the three Kiskundorozsma cemeteries. In the third (last) phase of the project, we will perform some statistical analyses (based on the macromorphological results) to estimate and compare the prevalence of leprosy in the three Kiskundorozsma series; and thus, to get a better insight into the past of the disease in the Hungarian Duna-Tisza Interfluve.

11) We were asked to " Please ensure that your manuscript meets PLOS ONE's style requirements, including those for file naming. The PLOS ONE style templates can be found at." We checked the style requirements of PLOS ONE, and it seems that our manuscript meets those requirements.

12) It was stated in the editorial letter that “In your Data Availability statement, you have not specified where the minimal data set underlying the results described in your manuscript can be found. PLOS defines a study's minimal data set as the underlying data used to reach the conclusions drawn in the manuscript and any additional data required to replicate the reported study findings in their entirety. All PLOS journals require that the minimal data set be made fully available. For more information about our data policy, please see http://journals.plos.org/plosone/s/data-availability.

We will update your Data Availability statement to reflect the information you provide in your cover letter.”

In the Data Availability Statement we provided when submitting the original version of our manuscript, it was already stated that “Yes - all data are fully available without restriction” and “All relevant data are within the manuscript and its Supporting Information files.” So, the “minimal data set underlying the results” can be found in our manuscript and its Supporting Information files. Our paper discusses only one case with leprosy (KK61) from a historic population – in the Ethics Statement, we already provided the exact geographical location of KK61 (i.e., Department of Biological Anthropology, University of Szeged, Közép fasor 52, H-6726 Szeged, Hungary), as well as the inventory number and grave number of KK61. All the observable skeletal remains of KK61 are presented in Figure 2B, all the applied macromorphological methods are mentioned in the Methods section (no statistical analyses were performed), and all the detected bony changes related to leprosy are described in the Results section and demonstrated in Figures 3–8. No other data are required to be “used to reach the conclusions drawn in the manuscript” and “replicate the reported study findings in their entirety”. In addition to the above, both reviewers answered “yes” to the question: “3. Have the authors made all data underlying the findings in their manuscript fully available? The PLOS Data policy requires authors to make all data underlying the findings described in their manuscript fully available without restriction, with rare exception (please refer to the Data Availability Statement in the manuscript PDF file). The data should be provided as part of the manuscript or its supporting information, or deposited to a public repository. For example, in addition to summary statistics, the data points behind means, medians and variance measures should be available. If there are restrictions on publicly sharing data—e.g. participant privacy or use of data from a third party—those must be specified.”.

Based on the above, we kindly ask the Editorial Office to check our Data Availability Statement once again, because we already provided all required information/data as part of the manuscript and its Supporting Information files (all information/data are fully available without restriction).

13) In the editorial letter, it was stated that “Please include your full ethics statement in the ‘Methods’ section of your manuscript file. In your statement, please include the full name of the IRB or ethics committee who approved or waived your study, as well as whether or not you obtained informed written or verbal consent. If consent was waived for your study, please include this information in your statement as well.”

In the originally submitted version of our manuscript, the full ethics statement (see below) was included to the Materials and Methods section as a subsection named “Ethics Statement” (between the Burial context subsection and the Methods subsection). In the revised version, it was moved to the end of the Methods section as a subsection:

“Ethics Statement

Specimen number: KK61 (inventory no. AP473; grave no. 61).

The skeleton evaluated in the described study is housed in the Department of Biological Anthropology, University of Szeged, in Szeged, Hungary. Access to the specimen was granted by the Department of Biological Anthropology, University of Szeged (Közép fasor 52, H-6726 Szeged, Hungary).

No permits were required for the described study, which complied with all relevant regulations.”

There was no IRB or ethics committee who should have approved or waived our study. Moreover, we did not have to obtain informed written or verbal consent as our paper is about a historic leprosy case (the individual have died hundreds of years ago). This is why this information is not provided in our Ethics Statement. Based on the above, we kindly ask the Editorial Office to check our Ethics Statement once again.

14) It was mentioned in the editorial letter that “We note that Figure 1 in your submission contain [map/satellite] images which may be copyrighted. All PLOS content is published under the Creative Commons Attribution License (CC BY 4.0), which means that the manuscript, images, and Supporting Information files will be freely available online, and any third party is permitted to access, download, copy, distribute, and use these materials in any way, even commercially, with proper attribution. For these reasons, we cannot publish previously copyrighted maps or satellite images created using proprietary data, such as Google software (Google Maps, Street View, and Earth). For more information, see our copyright guidelines: http://journals.plos.org/plosone/s/licenses-and-copyright.”

In Figure 1A, there is a map that we created using OpenStreetMap data. On the website of the PLOS ONE (https://journals.plos.org/plosone/s/figures), it was stated that “OpenStreetMap: OpenStreetMap map tiles are free to use as long as they are accompanied by the following attribution statement: “Base map and data from OpenStreetMap and OpenStreetMap Foundation”. Maps created using OpenStreetMap data must be accompanied by the following attribution statement: "Contains information from OpenStreetMap and OpenStreetMap Foundation, which is made available under the Open Database License.” Following these instructions, the legend of Figure 1 was supplemented with the attribution statement in the originally submitted version of our manuscript: “Contains information from OpenStreetMap and OpenStreetMap Foundation, which is made available under the Open Database License.” In the revised version of our manuscript, we slightly modified the legend to make it clear that it is Figure 1A that was created using OpenStreetMap data:

“Fig 1. A) Map of Hungary showing the location of the Kiskundorozsma–Kettőshatár I archaeological site; B) Aerial photo of the Kiskundorozsma–Kettőshatár I archaeological site; and C) Plan drawing of the late Avar Age cemetery of Kiskundorozsma–Kettőshatár I with the location of the burial of KK61. (Figure 1A contains information from OpenStreetMap and OpenStreetMap Foundation, which is made available under the Open Database License.)”

Based on the above, we think that we followed the requirements of PLOS ONE regarding copyrighted [map/satellite] images. Therefore, we kindly ask the Editorial Office to check Figure 1 and its legend once again.

15) In the editorial letter, we were asked to “Please review your reference list to ensure that it is complete and correct. If you have cited papers that have been retracted, please include the rationale for doing so in the manuscript text, or remove these references and replace them with relevant current references. Any changes to the reference list should be mentioned in the rebuttal letter that accompanies your revised manuscript. If you need to cite a retracted article, indicate the article’s retracted status in the References list and also include a citation and full reference for the retraction notice.”. We checked our reference list – it seems complete and correct; we could not find any retracted paper in it. 

In the revised version of our manuscript, we tried to execute all suggestions of the reviewers. I hope this new version will be suitable for publication in the PLOS ONE.

Thank you again for the reviewers’ insightful and constructive comments and your editorial work!

Yours sincerely,

Dr Olga Spekker, PhD

Postdoctoral researcher

Department of Biological Anthropology

University of Szeged

Közép fasor 52, H-6726 Szeged, Hungary

Email: olga.spekker@gmail.com

Tel: +36 20 807 72 94

---

## [Decision Letter · Decision Letter 1]

8 Feb 2022

Life and death of a leprosy sufferer from the 8th-century-CE cemetery of Kiskundorozsma–Kettőshatár I (Duna-Tisza Interfluve, Hungary) – Biological and social consequences of having Hansen’s disease in a late Avar Age population from Hungary

PONE-D-21-32338R1

Dear Dr. Spekker,

We’re pleased to inform you that your manuscript has been judged scientifically suitable for publication and will be formally accepted for publication once it meets all outstanding technical requirements.

Kind regards,

Mario Novak

Academic Editor

PLOS ONE

Additional Editor Comments (optional):

Reviewers' comments:

Reviewer's Responses to Questions

**Comments to the Author**

1. If the authors have adequately addressed your comments raised in a previous round of review and you feel that this manuscript is now acceptable for publication, you may indicate that here to bypass the “Comments to the Author” section, enter your conflict of interest statement in the “Confidential to Editor” section, and submit your "Accept" recommendation.

Reviewer #1: All comments have been addressed

Reviewer #2: All comments have been addressed

2. Is the manuscript technically sound, and do the data support the conclusions?

Reviewer #1: Yes

Reviewer #2: (No Response)

3. Has the statistical analysis been performed appropriately and rigorously? 

Reviewer #1: Yes

Reviewer #2: (No Response)

4. Have the authors made all data underlying the findings in their manuscript fully available?

Reviewer #1: Yes

Reviewer #2: (No Response)

5. Is the manuscript presented in an intelligible fashion and written in standard English?

Reviewer #1: Yes

Reviewer #2: (No Response)

6. Review Comments to the Author

Reviewer #1: All comments of the reviewers have been addressed and in my opinion the article can be published in PLOSOne

Reviewer #2: (No Response)

7. PLOS authors have the option to publish the peer review history of their article (what does this mean?). If published, this will include your full peer review and any attached files.

Reviewer #1: No

Reviewer #2: **Yes: **Željka Bedić

---

## [Editor Report · Acceptance letter]

10 Feb 2022

PONE-D-21-32338R1 

Life and death of a leprosy sufferer from the 8th-century-CE cemetery of Kiskundorozsma–Kettőshatár I (Duna-Tisza Interfluve, Hungary) – Biological and social consequences of having Hansen’s disease in a late Avar Age population from Hungary 

Dear Dr. Spekker:

I'm pleased to inform you that your manuscript has been deemed suitable for publication in PLOS ONE. Congratulations! Your manuscript is now with our production department. 

Kind regards, 

on behalf of

Dr. Mario Novak 

Academic Editor

PLOS ONE